# Robust Optimization for Non-Convex Objectives

**Robert Chen**
Computer Science
Harvard University

**Brendan Lucier**
Microsoft Research
New England

**Yaron Singer**
Computer Science
Harvard University

**Vasilis Syrgkanis**
Microsoft Research
New England

## Abstract

We consider robust optimization problems, where the goal is to optimize in the worst case over a class of objective functions. We develop a reduction from robust improper optimization to stochastic optimization: given an oracle that returns $\alpha$-approximate solutions for distributions over objectives, we compute a distribution over solutions that is $\alpha$-approximate in the worst case. We show that derandomizing this solution is NP-hard in general, but can be done for a broad class of statistical learning tasks. We apply our results to robust neural network training and submodular optimization. We evaluate our approach experimentally on corrupted character classification and robust influence maximization in networks.

## 1 Introduction

In many learning tasks we face uncertainty about the loss we aim to optimize. Consider, for example, a classification task such as character recognition, required to perform well under various types of distortion. In some environments, such as recognizing characters in photos, the classifier must handle rotation and patterned backgrounds. In a different environment, such as low-resolution images, it is more likely to encounter noisy pixelation artifacts. Instead of training a separate classifier for each possible scenario, one seeks to optimize performance in the worst case over different forms of corruption (or combinations thereof) made available to the trainer as black-boxes.

More generally, our goal is to find a minimax solution that optimizes in the worst case over a given family of functions. Even if each individual function can be optimized effectively, it is not clear such solutions would perform well in the worst case. In many cases of interest, individual objectives are non-convex and hence state-of-the-art methods are only approximate. In stochastic optimization, where one must optimize a distribution over loss functions, approximate stochastic optimization is often straightforward, since loss functions are commonly closed under convex combination. Can approximately optimal stochastic solutions yield an approximately optimal *robust* solution?

In this paper we develop a reduction from robust optimization to stochastic optimization. Given an $\alpha$-approximate oracle for stochastic optimization we show how to implement an $\alpha$-approximate solution for robust optimization under a necessary extension, and illustrate its effectiveness in applications.

**Main Results.** Given an $\alpha$-approximate stochastic oracle for distributions over (potentially non-convex) loss functions, we show how to solve $\alpha$-approximate robust optimization in a convexified solution space. This outcome is "improper" in the sense that it may lie outside the original solution space, if the space is non-convex. This can be interpreted as computing a distribution over solutions. We show that the relaxation to improper learning is necessary in general: It is NP-hard to achieve robust optimization with respect to the original outcome space, even if stochastic optimization can be solved exactly, and even if there are only polynomially many loss functions. We complement this by showing that in any statistical learning scenario where loss is convex in the predicted dependent variable, we can find a single (deterministic) solution with matching performance guarantees.

**Technical overview.** Our approach employs an execution of no-regret dynamics on a zero-sum game, played between a learner equipped with an $\alpha$-approximate stochastic oracle, and an adversary who aims to find a distribution over loss functions that maximizes the learner's loss. This game converges to an approximately robust solution, in which the learner and adversary settle upon an $\alpha$-approximate minimax solution. This convergence is subject to an additive regret term that converges at a rate of $T^{-1/2}$ over $T$ rounds of the learning dynamics.

**Applications.** We illustrate the power of our reduction through two main examples. We first consider statistical learning via neural networks. Given an arbitrary training method, our reduction generates a net that optimizes robustly over a given class of loss functions. We evaluate our method experimentally on a character recognition task, where the loss functions correspond to different corruption models made available to the learner as black boxes. We verify experimentally that our approach significantly outperforms various baselines, including optimizing for average performance and optimizing for each loss separately. We also apply our reduction to influence maximization, where the goal is to maximize a concave function (the independent cascade model of influence [11]) over a non-convex space (subsets of vertices in a network). Previous work has studied robust influence maximization directly [9, 5, 15], focusing on particular, natural classes of functions (e.g., edge weights chosen within a given range) and establishing hardness and approximation results. In comparison, our method is agnostic to the particular class of functions, and achieves a strong approximation result by returning a distribution over solutions. We evaluate our method on real and synthetic datasets, with the goal of robustly optimizing a suite of random influence instantiations. We verify experimentally that our approach significantly outperforms natural baselines.

**Related work.** There has recently been a great deal of interest in robust optimization in machine learning [20, 4, 17, 21, 16]. For continuous optimization, the work that is closest to ours is perhaps that by Shalev-Shwartz and Wexler [20] and Namkoong and Duchi [17] that use robust optimization to train against convex loss functions. The main difference is that we assume a more general setting in which the loss functions are non-convex and one is only given access to the stochastic oracle. Hence, the proof techniques and general results from these papers do not apply to our setting. We note that our result generalizes these works, as they can be considered as the special case in which we have a distributional oracle whose approximation is optimal. In particular, [20, Theorem 1] applies to the realizable statistical learning setting where the oracle has small mistake bound C. Our applications require a more general framing that hold for any optimization setting with access to an approximate oracle, and approximation is in the multiplicative sense with respect to the optimal value. In submodular optimization there has been a great deal of interest in robust optimization as well [12, 13, 10, 6]. The work closest to ours is that by He and Kempe [10] who consider a slightly different objective than ours. Kempe and He's results apply to influence but do not extend to general submodular functions. Finally, we note that unlike recent work on non-convex optimization [7, 1, 8] our goal in this paper is not to optimize a non-convex function. Rather, we abstract the non-convex guarantees via the approximate stochastic oracle.

## 2 Robust Optimization with Approximate Stochastic Oracles

We consider the following model of optimization that is robust to objective uncertainty. There is a space $\mathcal{X}$ over which to optimize, and a finite set of loss functions[1] $\mathcal{L} = \{L_1, \ldots, L_m\}$ where each $L_i \in \mathcal{L}$ is a function from $\mathcal{X}$ to $[0, 1]$. Intuitively, our goal is to find some $x \in \mathcal{X}$ that achieves low loss in the worst-case over loss functions in $\mathcal{L}$. For $x \in \mathcal{X}$, write $g(x) = \max_{i \in [m]} L_i(x)$ for the worst-case loss of $x$. The minimax optimum $\tau$ is given by

$$\tau = \min_{x \in \mathcal{X}} g(x) = \min_{x \in \mathcal{X}} \max_{i \in [m]} L_i(x). \tag{1}$$

The goal of $\alpha$-approximate robust optimization is to find $x$ such that $g(x) \leq \alpha\tau$.[2]

**Algorithm 1** Oracle Efficient Improper Robust Optimization
---
**Input:** Objectives $\mathcal{L} = \{L_1, \ldots, L_m\}$, Apx stochastic oracle $M$, parameters $T, \eta$
**for** each time step $t \in [T]$ **do**
    Set

$$\mathbf{w}_t[i] \propto \exp\left\{\eta \sum_{\tau=1}^{t-1} L_i(x_\tau)\right\} \qquad (3)$$

    Set $x_t = M(\mathbf{w}_t)$
**end for**
**Output:** the uniform distribution over $\{x_1, \ldots, x_T\}$

---

Given a distribution $\mathcal{P}$ over solutions $\mathcal{X}$, write $g(\mathcal{P}) = \max_{i \in [m]} \mathbb{E}_{x \sim \mathcal{P}}[L_i(x)]$ for the worst-case expected loss of a solution drawn from $\mathcal{P}$. A weaker version of robust approximation is *improper robust optimization*: find a distribution $\mathcal{P}$ over $\mathcal{X}$ such that $g(\mathcal{P}) \leq \alpha\tau$.

Our results take the form of reductions to an approximate stochastic oracle, which finds a solution $x \in \mathcal{X}$ that approximately minimizes a given distribution over loss functions.[3]

**Definition 1** ($\alpha$-Approximate Stochastic Oracle). *Given a distribution $D$ over $\mathcal{L}$, an $\alpha$-approximate stochastic oracle $M(D)$ computes $x^* \in \mathcal{X}$ such that*

$$\mathbb{E}_{L \sim D}[L(x^*)] \leq \alpha \min_{x \in \mathcal{X}} \mathbb{E}_{L \sim D}[L(x)]. \qquad (2)$$

### 2.1 Improper Robust Optimization with Oracles

We first show that, given access to an $\alpha$-approximate stochastic oracle, it is possible to efficiently implement improper $\alpha$-approximate robust optimization, subject to a vanishing additive loss term.

**Theorem 1.** *Given access to an $\alpha$-approximate stochastic oracle, Algorithm 1 with $\eta = \sqrt{\frac{\log(m)}{2T}}$ computes a distribution $\mathcal{P}$ over solutions, defined as a uniform distribution over a set $\{x_1, \ldots, x_T\}$, so that*

$$\max_{i \in [m]} \mathbb{E}_{x \sim \mathcal{P}}[L_i(x)] \leq \alpha\tau + \sqrt{\frac{2\log(m)}{T}}. \qquad (4)$$

*Moreover, for any $\eta$ the distribution $\mathcal{P}$ computed by Algorithm 1 satisfies:*

$$\max_{i \in [m]} \mathbb{E}_{x \sim \mathcal{P}}[L_i(x)] \leq \alpha(1+\eta)\tau + \frac{\log(m)}{\eta T}. \qquad (5)$$

*Proof.* We give the proof of the first result and defer the second result to the full version of the paper. We can interpret Algorithm 1 in the following way. We define a zero-sum game between a learner and an adversary. The learner's action set is equal to $\mathcal{X}$ and the adversary's action set is equal to $[m]$. The loss of the learner when he picks $x \in \mathcal{X}$ and the adversary picks $i \in [m]$ is defined as $L_i(x)$. The corresponding payoff of the adversary is $L_i(x)$.

We will run no-regret dynamics on this zero-sum game, where at every iteration $t = 1, \ldots, T$, the adversary will pick a distribution over functions and subsequently the learner picks a solution $x_t$. For simpler notation we will denote with $\mathbf{w}_t$ the probability density function on $[m]$ associated with the distribution of the adversary. That is, $w_t[i]$ is the probability of picking function $L_i \in \mathcal{L}$. The adversary picks a distribution $\mathbf{w}_t$ based on some arbitrary no-regret learning algorithm on the $m$ actions in $\mathcal{L}$. For concreteness consider the case where the adversary picks a distribution based on the multiplicative weight updates algorithm, i.e.,

$$w_t[i] \propto \exp\left\{\sqrt{\frac{\log(m)}{2T}} \sum_{\tau=1}^{t-1} L_i(x_\tau)\right\}. \qquad (6)$$

Subsequently the learner picks a solution $x_t$ that is the output of the $\alpha$-approximate stochastic oracle on the distribution selected by the adversary at time-step $t$. That is,

$$x_t = M\left(\mathbf{w}_t\right). \tag{7}$$

Write $\epsilon(T) = \sqrt{\frac{2\log(m)}{T}}$. By the guarantees of the no-regret algorithm for the adversary, we have that

$$\frac{1}{T}\sum_{t=1}^{T}\mathbb{E}_{I\sim\mathbf{w}_t}\left[L_I(x_t)\right] \geq \max_{i\in[m]}\frac{1}{T}\sum_{t=1}^{T}L_i(x_t) - \epsilon(T). \tag{8}$$

Combining the above with the guarantee of the stochastic oracle we have

$$\tau = \min_{x\in\mathcal{X}}\max_{i\in[m]}L_i(x) \geq \min_{x\in X}\frac{1}{T}\sum_{t=1}^{T}\mathbb{E}_{I\sim\mathbf{w}_t}\left[L_I(x)\right] \geq \frac{1}{T}\sum_{t=1}^{T}\min_{x\in X}\mathbb{E}_{I\sim\mathbf{w}_t}\left[L_I(x)\right]$$

$$\geq \frac{1}{T}\sum_{t=1}^{T}\frac{1}{\alpha}\cdot\mathbb{E}_{I\sim\mathbf{w}_t}\left[L_I(x_t)\right] \qquad \text{(By oracle guarantee for each } t\text{)}$$

$$\geq \frac{1}{\alpha}\cdot\left(\max_{i\in[m]}\frac{1}{T}\sum_{t=1}^{T}L_i(x_t) - \epsilon(T)\right). \qquad \text{(By no-regret of adversary)}$$

Thus, if we define with $\mathcal{P}$ to be the uniform distribution over $\{x_1,\ldots,x_T\}$, then we have derived

$$\max_{i\in[m]}\mathbb{E}_{x\sim\mathcal{P}}\left[L_i(x)\right] \leq \alpha\tau + \epsilon(T) \tag{9}$$

as required. $\qquad\square$

A corollary of Theorem 1 is that if the solution space $\mathcal{X}$ is convex and the objective functions $L_i \in \mathcal{L}$ are all convex functions, then we can compute a single solution $x^*$ that is approximately minimax optimal. Of course, in this setting one can calculate and optimize the maximum loss directly in time proportional to $|\mathcal{L}|$; this result therefore has the most bite when the set of functions is large.

**Corollary 2.** *If the space $\mathcal{X}$ is a convex space and each loss function $L_i \in \mathcal{L}$ is a convex function, then the point $x^* = \frac{1}{T}\sum_{t=1}^{T}x_t \in \mathcal{X}$, where $\{x_1,\ldots,x_T\}$ are the output of Algorithm 1, satisfies:*

$$\max_{i\in[m]}L_i(x^*) \leq \alpha\tau + \sqrt{\frac{2\log(m)}{T}} \tag{10}$$

*Proof.* By Theorem 1, we get that if $\mathcal{P}$ is the uniform distribution over $\{x_1,\ldots,x_T\}$ then

$$\max_{i\in[m]}\mathbb{E}_{x\sim\mathcal{P}}[L_i(x)] \leq \alpha\tau + \sqrt{\frac{2\log(m)}{T}}.$$

Since $\mathcal{X}$ is convex, the solution $x^* = \mathbb{E}_{x\sim\mathcal{P}}[x]$ is also part of $\mathcal{X}$. Moreover, since each $L_i \in \mathcal{L}$ is convex, we have that $\mathbb{E}_{x\sim\mathcal{P}}[L_i(x)] \geq L_i(\mathbb{E}_{x\sim\mathcal{P}}[x]) = L_i(x^*)$. We therefore conclude

$$\max_{i\in[m]}L_i(x^*) \leq \max_{i\in[m]}\mathbb{E}_{x\sim\mathcal{P}}[L_i(x)] \leq \alpha\tau + \sqrt{\frac{2\log(m)}{T}}$$

as required. $\qquad\square$

## 2.2 Robust Statistical Learning

Next we apply our main theorem to statistical learning. Consider regression or classification settings where data points are pairs $(z,y)$, $z \in \mathcal{Z}$ is a vector of features, and $y \in \mathcal{Y}$ is the dependent variable. The solution space $\mathcal{X}$ is then a space of hypotheses $\mathcal{H}$, with each $h \in \mathcal{H}$ a function from $\mathcal{Z}$ to $\mathcal{Y}$. We also assume that $\mathcal{Y}$ is a convex subset of a finite-dimensional vector space.

We are given a set of loss functions $\mathcal{L} = \{L_1,\ldots,L_m\}$, where each $L_i \in \mathcal{L}$ is a functional $L_i\colon \mathcal{H} \to [0,1]$. Theorem 1 implies that, given an $\alpha$-approximate stochastic optimization oracle,

we can compute a distribution over $T$ hypotheses from $\mathcal{H}$ that achieves an $\alpha$-approximate minimax guarantee. If the loss functionals are convex over hypotheses, then we can compute a single ensemble hypothesis $h^*$ (possibly from a larger space of hypotheses, if $\mathcal{H}$ is non-convex) that achieves this guarantee.

**Theorem 3.** *Suppose that $\mathcal{L} = \{L_1, \dots, L_m\}$ are convex functionals. Then the ensemble hypothesis $h^* = \frac{1}{T}\sum_{t=1}^{T} h$, where $\{h_1, \dots, h_T\}$ are the hypotheses output by Algorithm 1 given an $\alpha$-approximate stochastic oracle, satisfies*

$$\max_{i \in [m]} L_i(h^*) \leq \alpha \min_{h \in H} \max_{i \in [m]} L_i(h) + \sqrt{\frac{2\log(m)}{T}}. \tag{11}$$

*Proof.* The proof is similar to the proof of Corollary 2. $\qquad\square$

We emphasize that the convexity condition in Theorem 3 is over the class of hypotheses, rather than over features or any natural parameterization of $\mathcal{H}$ (such as weights in a neural network). This is a mild condition that applies to many examples in statistical learning theory. For instance, consider the case where each loss $L_i(h)$ is the expected value of some ex-post loss function $\ell_i(h(z), y)$ given a distribution $D_i$ over $Z \times Y$:

$$L_i(h) = \mathbb{E}_{(z,y) \sim D_i}\left[\ell_i(h(z), y)\right]. \tag{12}$$

In this case, it is enough for the function $\ell_i(\cdot, \cdot)$ to be convex with respect to its first argument (i.e., the predicted dependent variable). This is satisfied by most loss functions used in machine learning, such as multinomial logistic loss (cross-entropy loss) $\ell(\hat{y}, y) = -\sum_{c \in [k]} y_c \log(\hat{y}_c)$ from multi-class classification or squared loss $\ell(\hat{y}, y) = \|\hat{y} - y\|^2$ as used in regression. For all these settings, Theorem 3 provides a tool for improper robust learning, where the final hypothesis $h^*$ is an ensemble of $T$ base hypotheses from $\mathcal{H}$. Again, the underlying optimization problem can be arbitrarily non-convex in the natural parameters of the hypothesis space; in Section 3.1 we will show how to apply this approach to robust training of neural networks, where the stochastic oracle is simply a standard network training method. For neural networks, the fact that we achieve improper learning (as opposed to standard learning) corresponds to training a neural network with a single extra layer relative to the networks generated by the oracle.

## 2.3 Robust Submodular Maximization

In *robust submodular maximization* we are given a family of reward functions $\mathcal{F} = \{f_1, \dots, f_m\}$, where each $f_i \in \mathcal{F}$ is a monotone submodular function from a ground set $N$ of $n$ elements to $[0, 1]$. Each function is assumed to be monotone and submodular, i.e., for any $S \subseteq T \subseteq N$, $f_i(S) \leq f_i(T)$; and for any $S, T \subseteq N$, $f(S \cup T) + f(S \cap T) \leq f(S) + f(T)$. The goal is to select a set $S \subseteq N$ of size $k$ whose worst-case value over $i$, i.e., $g(S) = \min_{i \in [m]} f_i(S)$, is at least a $1/\alpha$ factor of the minimax optimum $\tau = \max_{T:|T| \leq k} \min_{i \in [m]} f_i(T)$.

This setting is a special case of our general robust optimization setting (phrased in terms of rewards rather than losses). The solution space $\mathcal{X}$ is equal to the set of subsets of size $k$ among all elements in $N$ and the set $\mathcal{F}$ is the set of possible objective functions. The stochastic oracle 1, instantiated in this setting, asks for the following: given a convex combination of submodular functions $F(S) = \sum_{i=1}^{m} \mathbf{w}[i] \cdot f_i(S)$, compute a set $S^*$ such that $F(S^*) \geq \frac{1}{\alpha} \max_{S:|S| \leq k} F(S)$.

Computing the maximum value set of size $k$ is NP-hard even for a single submodular function. The following very simple greedy algorithm computes a $(1 - 1/e)$-approximate solution [19]: begin with $S_{cur} = \emptyset$, and at each iteration add to the current solution $S_{cur}$ the element $j \in N - S_{cur}$ that has the largest marginal contribution: $f(\{j\} \cup S_{cur}) - f(S_{cur})$. Moreover, this approximation ratio is known to be the best possible in polynomial time [18]. Since a convex combination of monotone submodular functions is also a monotone submodular function, we immediately get that there exists a $(1 - 1/e)$-approximate stochastic oracle that can be computed in polynomial time. The algorithm is formally given in Algorithm 2. Combining the above with Theorem 1 we get the following corollary.

**Corollary 4.** *Algorithm 1, with stochastic oracle $M_{greedy}$, computes in time $\mathrm{poly}(T, n)$ a distribution $\mathcal{P}$ over sets of size $k$, defined as a uniform distribution over a set $\{S_1, \dots, S_T\}$, such that*

$$\min_{i \in [m]} \mathbb{E}_{S \sim \mathcal{P}}\left[f_i(S)\right] \geq \left(1 - \frac{1}{e}\right)(1 - \eta)\tau - \frac{\log(m)}{\eta T}. \tag{13}$$

**Algorithm 2** Greedy stochastic Oracle for Submodular Maximization $M_{greedy}$

---

**Input:** Set of elements $N$, objectives $\mathcal{F} = \{f_1, \dots, f_m\}$, distribution over objectives $\mathbf{w}$
Set $S_{cur} = \emptyset$
**for** $j = 1$ to $k$ **do**
    Let $j^* = \arg\max_{j \in N - S_{cur}} \sum_{i=1}^{m} \mathbf{w}[i] \left( f_i(\{j\} \cup S_{cur}) - f_i(S_{cur}) \right)$
    Set $S_{cur} = \{j^*\} \cup S_{cur}$
**end for**

---

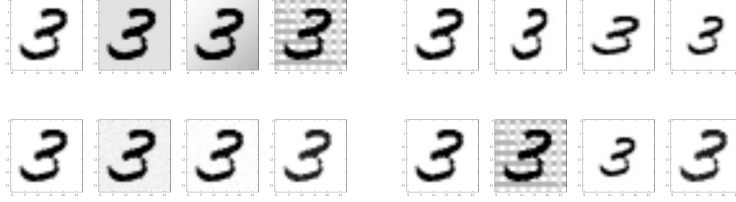

Figure 1: Sample MNIST image with each of the corruptions applied to it. Background Corruption Set & Shrink Corruption Set (top). Pixel Corruption Set & Mixed Corruption Set (bottom).

We show in the full version of the paper that computing a single set $S$ that achieves a $(1 - 1/e)$-approximation to $\tau$ is also $NP$-hard. This is true even if the functions $f_i$ are additive. However, by allowing a randomized solution over sets we can achieve a constant factor approximation to $\tau$ in polynomial time.

Since the functions are monotone, the above result implies a simple way of constructing a single set $S^*$ that is of larger size than $k$, which deterministically achieves a constant factor approximation to $\tau$. The latter holds by simply taking the union of the sets $\{S_1, \dots, S_T\}$ in the support of the distribution returned by Algorithm 1. We get the following bi-criterion approximation scheme.

**Corollary 5.** *Suppose that we run the reward version of Algorithm 1 with $\eta = \epsilon$ and for $T = \frac{\log(m)}{\tau \epsilon^2}$, returning $\{S_1, \dots, S_T\}$. Then the set $S^* = S_1 \cup \dots \cup S_T$, which is of size at most $\frac{k \log(m)}{\tau \epsilon^2}$, satisfies*

$$\min_{i \in [m]} f_i(S^*) \geq \left( 1 - \frac{1}{e} - 2\epsilon \right) \tau. \tag{14}$$

# 3 Experiments[4]

## 3.1 Robust Classification with Neural Networks

A classic application of our robust optimization framework is classification with neural networks for corrupted or perturbed datasets. We have a data set $Z$ of pairs $(z, y)$ of an image $z \in \mathcal{Z}$ and label $y \in \mathcal{Y}$ that can be corrupted in $m$ different ways which produces data sets $Z_1, \dots, Z_m$. The hypothesis space $H$ is the set of all neural nets of some fixed architecture and for each possible assignment of weights. We denote each such hypothesis with $h(\cdot; \theta) : \mathcal{Z} \to \mathcal{Y}$ for $\theta \in \mathbb{R}^d$, with $d$ being the number of parameters (weights) of the neural net. If we let $D_i$ be the uniform distribution over each corrupted data set $Z_i$, then we are interested in minimizing the empirical cross-entropy (aka multinomial logistic) loss in the worst case over these different distributions $D_i$. The latter is a special case of our robust statistical learning framework from Section 2.2.

Training a neural network is a non-convex optimization problem and we have no guarantees on its performance. We instead assume that for any given distribution $D$ over pairs $(z, y)$ of images and labels and for any loss function $\ell(h(z; \theta), y)$, training a neural net with *stochastic gradient descent* run on images drawn from $D$ can achieve an $\alpha$ approximation to the optimal expected loss, i.e. $\min_{\theta \in \mathbb{R}^d} \mathbb{E}_{(z,y) \sim D} \left[ \ell(h(z; \theta), y) \right]$. Notice that this implies an $\alpha$-approximate stochastic oracle for the

corrupted dataset robust training problem: for any distribution $\mathbf{w}$ over the different corruptions $[m]$, the stochastic oracle asks to give an $\alpha$-approximation to the minimization problem:

$$\min_{\theta \in \mathbb{R}^d} \sum_{i=1}^{m} \mathbf{w}[i] \cdot \mathbb{E}_{(z,y) \sim D_i} \left[ \ell(h(z;\theta), y) \right] \tag{15}$$

The latter is simply another expected loss problem with distribution over images being the mixture distribution defined by first drawing a corruption index $i$ from $\mathbf{w}$ and then drawing a corrupted image from distribution $D_i$. Hence, our oracle assumption implies that SGD on this mixture is an $\alpha$-approximation. By linearity of expectation, an alternative way of viewing the stochastic oracle problem is that we are training a neural net on the original distribution of images, but with loss function being the weighted combination of loss functions $\sum_{i=1}^{m} \mathbf{w}[i] \cdot \ell(h(c_i(z);\theta), y)$, where $c_i(z)$ is the $i$-th corrupted version of image $z$. In our experiments we implemented both of these interpretations of the stochastic oracle, which we call the *Hybrid Method* and *Composite Method*, respectively, when designing our neural network training scheme (see the full version of the paper for further details). Finally, because we use the cross-entropy loss, which is convex in the prediction of the neural net, we can also apply Theorem 3 to get that the ensemble neural net, which takes the average of the predictions of the neural nets created at each iteration of the robust optimization, will also achieve good worst-case loss (we refer to this as *Ensemble Bottleneck Loss*).

**Experiment Setup.** We use the MNIST handwritten digits data set containing 55000 training images, 5000 validation images, and 10000 test images, each image being a $28 \times 28$ pixel grayscale image. The intensities of these 576 pixels (ranging from 0 to 1) are used as input to a neural network that has 1024 nodes in its one hidden layer. The output layer uses the softmax function to give a distribution over digits 0 to 9. The activation function is ReLU and the network is trained using Gradient Descent with learning parameter 0.5 through 500 iterations of mini-batches of size 100.

In general, the corruptions can be any black-box corruption of the image. In our experiments, we consider four types of corruption ($m = 4$). See the full version of the paper for further details about corruptions.

**Baselines.** We consider three baselines: (i) *Individual Corruption*: for each corruption type $i \in [m]$, we construct an oracle that trains a neural network using the training data perturbed by corruption $i$, and then returns the trained network weights as $\theta_t$, for every $t = 1, \ldots, T$. This gives $m$ baselines, one for each corruption type; (ii) *Even Split*: this baseline alternates between training with different corruption types between iterations. In particular, call the previous $m$ baseline oracles $O_1, ..., O_m$. Then this new baseline oracle will produce $\theta_t$ with $O_{i+1}$, where $i \equiv t \mod m$, for every $t = 1, ..., T$; (iii) *Uniform Distribution*: This more advanced baseline runs the robust optimization scheme with the Hybrid Method (see Appendix), but without the distribution updates. Instead, the distribution over corruption types is fixed as the discrete uniform $[\frac{1}{m}, ..., \frac{1}{m}]$ over all $T$ iterations. This allows us to check if the multiplicative weight updates in the robust optimization algorithm are providing benefit.

**Results.** The Hybrid and Composite Methods produce results far superior to all three baseline types, with differences both substantial in magnitude and statistically significant, as shown in Figure 2. The more sophisticated Composite Method outperforms the Hybrid Method. Increasing $T$ improves performance, but with diminishing returns–largely because for sufficiently large $T$, the distribution over corruption types has moved from the initial uniform distribution to some more optimal *stable distribution* (see the full version for details). All these effects are consistent across the 4 different corruption sets tested. The Ensemble Bottleneck Loss is empirically much smaller than Individual Bottleneck Loss. For the best performing algorithm, the Composite Method, the mean Ensemble Bottleneck Loss (mean Individual Bottleneck Loss) with $T = 50$ was 0.34 (1.31) for Background Set, 0.28 (1.30) for Shrink Set, 0.19 (1.25) for Pixel Set, and 0.33 (1.25) for Mixed Set. Thus combining the $T$ classifiers obtained from robust optimization is practical for making predictions on new data.

## 3.2 Robust Influence Maximization

We apply the results of Section 2.3 to the robust influence maximization problem. Given a directed graph $G = (V, E)$, the goal is to pick a *seed set* $S$ of $k$ nodes that maximize an influence function

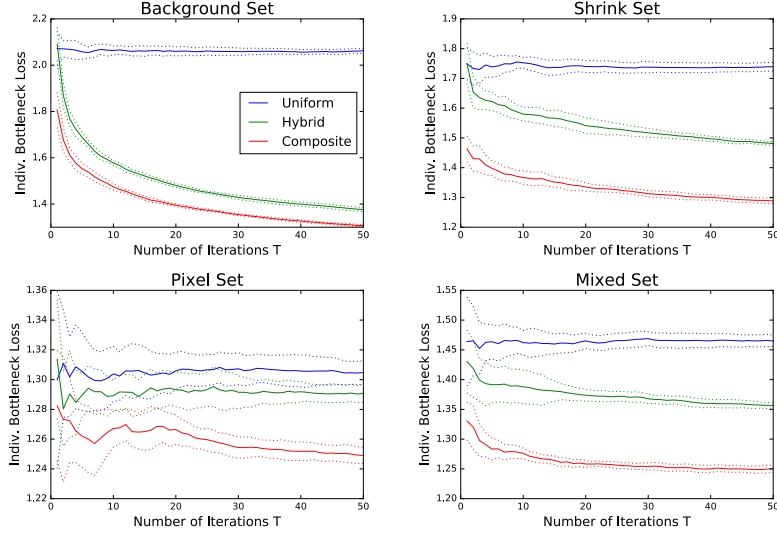

Figure 2: Comparison of methods, showing mean of 10 independent runs and a 95% confidence band. The criterion is *Individual Bottleneck Loss*: $\min_{[m]} E_{\theta \sim P} [\ell(h(z; \theta), y)]$, where $P$ is uniform over all solutions $\theta_i$ for that method. Baselines (i) and (ii) are not shown as they produce significantly higher loss.

$f_G(S)$, where $f_G(S)$ is the expected number of individuals influenced by opinion of the members of $S$. We used $f_G(S)$ to be the number of nodes reachable from $S$ (our results extend to other models).

In robust influence maximization, the goal is to maximize influence in the worst-case (*Bottleneck Influence*) over $m$ functions $\{f_1, \ldots, f_m\}$, corresponding to $m$ graphs $\{G_1, \ldots, G_m\}$, for some fixed seed set of size $k$. This is a special case of robust submodular maximization after rescaling to $[0, 1]$.

**Experiment Setup.** Given a base directed graph $G(V, E)$, we produce $m$ graphs $G_i = (V, E_i)$ by randomly including each edge $e \in E$ with some probability $p$. We consider two base graphs and two sets of parameters for each: (i) The *Wikipedia Vote Graph* [14]. In Experiment $A$, the parameters are $|V| = 7115$, $|E| = 103689$, $m = 10$, $p = 0.01$ and $k = 10$. In Experiment $B$, change $p = 0.015$ and $k = 3$. (ii) The *Complete Directed Graph* on $|V| = 100$ vertices. In Experiment $A$, the parameters are $m = 50$, $p = 0.015$ and $k = 2$. In Experiment $B$, change $p = 0.01$ and $k = 4$.

**Baselines.** We compared our algorithm (Section 2.3) to three baselines: (i) *Uniform over Individual Greedy Solutions*: Apply greedy maximization (Algorithm 2) on each graph separately, to get solutions $\{S_1^g, \ldots, S_m^g\}$. Return the uniform distribution over these solutions; (ii) *Greedy on Uniform Distribution over Graphs*: Return the output of greedy submodular maximization (Algorithm 2) on the uniform distribution over influence functions. This can be viewed as maximizing expected influence; (iii) *Uniform over Greedy Solutions on Multiple Perturbed Distributions*: Generate $T$ distributions $\{\mathbf{w}_1^*, \ldots, \mathbf{w}_T^*\}$ over the $m$ functions, by randomly perturbing the uniform distribution. Perturbation magnitudes were chosen s.t. $\mathbf{w}_t^*$ has the same expected $\ell_1$ distance from uniform as the distribution returned by robust optimization at iteration $t$.

**Results.** For both graph experiments, robust optimization outperforms all baselines on Bottleneck Influence; the difference is statistically significant as well as large in magnitude for all $T > 50$ (see Figure 3). Moreover, the individual seed sets generated at each iteration $t$ of robust optimization themselves achieve empirically good influence as well; see the full version for further details.

## Footnotes

[1] We describe an extension to infinite sets of loss functions in the full version of the paper. Our results also extend naturally to the goal of maximizing the minimum of a class of reward functions.

[2] This oracle framework is similar to that used by Ben-Tal et al. [3], but where the approximation is multiplicative rather than additive.

[3]All our results easily extend to the case where the oracle computes a solution that is approximately optimal up to an additive error, rather than a multiplicative one. For simplicity of exposition we present the multiplicative error case as it is more in line with the literature on approximation algorithms.

[4]Code used to implement the algorithms and run the experiments is available at `https://github.com/12degrees/Robust-Classification/`.

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
