[Supplementary Material]

# Supplementary material for
# "Robust Optimization for Non-Convex Objectives"

## A  Faster Convergence to Approximate Solution

**Theorem 6** (Faster Convergence). *Given access to an $\alpha$-approximate stochastic oracle, Algorithm 1 with some parameter $\eta$ computes a distribution $\mathcal{P}$ over solutions, defined as a uniform distribution over a set $\{x_1, \ldots, x_T\}$, such that*

$$\max_{i \in [m]} \mathbb{E}_{x \sim \mathcal{P}} \left[ L_i(x) \right] \leq \alpha(1 + \eta)\tau + \frac{\log(m)}{\eta T}. \tag{16}$$

*In the case of robust reward maximization, the reward version of Algorithm 1 computes a distribution $\mathcal{P}$ such that:*

$$\min_{i \in [m]} \mathbb{E}_{x \sim \mathcal{P}} \left[ L_i(x) \right] \geq \alpha(1 - \eta)\tau - \frac{\log(m)}{\eta T}. \tag{17}$$

*Proof.* We present the case of losses as the result for the case of rewards follows along similar lines. The proof follows similar lines as that of Theorem 1. The main difference is that we use a stronger property of the Exponential Weight Updates algorithm. In particular it is known that the regret of EWU, when run on a sequence of rewards that lie in $[-1, 1]$ is at most [2]:

$$\epsilon(T) = \eta \sum_{t=1}^{T} \mathbb{E}_{I \sim \mathbf{w}_t} \left[ L_I(x_t)^2 \right] + \frac{\log(m)}{\eta T} \leq \eta \sum_{t=1}^{T} \mathbb{E}_{I \sim \mathbf{w}_t} \left[ L_I(x_t) \right] + \frac{\log(m)}{\eta T} \tag{18}$$

where the second inequality follows from the fact that $L_i(x) \in [0, 1]$. Thus, by the definition of regret, we can write:

$$(1 + \eta) \frac{1}{T} \sum_{t=1}^{T} \mathbb{E}_{I \sim \mathbf{w}_t} \left[ L_I(x_t) \right] \geq \max_{i \in [m]} \frac{1}{T} \sum_{t=1}^{T} L_i(x_t) - \frac{\log(m)}{\eta T}. \tag{19}$$

Combining the above with the guarantee of the stochastic oracle we have

$$
\begin{aligned}
\tau = \min_{x \in \mathcal{X}} \max_{i \in [m]} L_i(x) \geq\ & \min_{x \in X} \frac{1}{T} \sum_{t=1}^{T} \mathbb{E}_{I \sim \mathbf{w}_t} \left[ L_I(x) \right] \\
\geq\ & \frac{1}{T} \sum_{t=1}^{T} \min_{x \in X} \mathbb{E}_{I \sim \mathbf{w}_t} \left[ L_I(x) \right] \\
\geq\ & \frac{1}{T} \sum_{t=1}^{T} \frac{1}{\alpha} \cdot \mathbb{E}_{I \sim \mathbf{w}_t} \left[ L_I(x_t) \right] && \text{(By oracle guarantee for each } t) \\
\geq\ & \frac{1}{\alpha(1 + \eta)} \cdot \left( \max_{i \in [m]} \frac{1}{T} \sum_{t=1}^{T} L_i(x_t) - \frac{\log(m)}{\eta T} \right). && \text{(By regret of adversary)}
\end{aligned}
$$

Thus, if we define with $\mathcal{P}$ to be the uniform distribution over $\{x_1, \ldots, x_T\}$, then we have derived:

$$\max_{i \in [m]} \mathbb{E}_{x \sim \mathcal{P}} \left[ L_i(x) \right] \leq \alpha(1 + \eta)\tau + \frac{\log(m)}{\eta T} \tag{20}$$

as required. □

## B  Robust Optimization with Infinite Loss Sets

We now extend our main results to the case where the uncertainty about the loss function is more general. In particular, we allow for sets of possible losses $\mathcal{L}$ that are not necessary finite. In particular, the loss function depends on a parameter $w \in \mathcal{W}$ that is unknown and which could take any value in

---

**Algorithm 3** Oracle Efficient Improper Robust Optimization with Infinite Loss Sets

---
**Input:** A convex set $\mathcal{Y}$ and loss function $L(\cdot, \cdot)$ which defines the set of possible losses $\mathcal{L}$
**Input:** Approximately optimal stochastic oracle $M$
**Input:** Accuracy parameter $T$ and step-size $\eta$
**for** each time step $t \in [T]$ **do**
    Set

$$\theta_t = \theta_{t-1} + \nabla_y L(x_t, y_t) \tag{24}$$
$$w_t = \Pi_{\mathcal{W}} (\eta \cdot \theta_t) \tag{25}$$

    Set $x_t = M(w_t)$
**end for**
Output the uniform distribution over $\{x_1, \ldots, x_T\}$

---

a set $\mathcal{W}$. The loss of the learner is a function $L(x, w)$ of both his action $x \in \mathcal{X}$ and this parameter $w \in \mathcal{W}$, and the form of the function $L$ is known. Hence, the set of possible losses is defined as:

$$\mathcal{L} = \{L(\cdot, w) : w \in \mathcal{W}\} \tag{21}$$

Our goal is to find some $x \in \mathcal{X}$ that achieves low loss in the worst-case over loss functions in $\mathcal{L}$. For $x \in \mathcal{X}$, write $g(x) = \max_{w \in \mathcal{W}} L(x, w)$ for the worst-case loss of $x$. The minimax optimum is

$$\tau = \min_{x \in \mathcal{X}} g(x) = \min_{x \in \mathcal{X}} \max_{w \in \mathcal{W}} L(x, w). \tag{22}$$

Our goal in $\alpha$-approximate robust optimization is to find $x$ such that $g(x) \leq \alpha\tau$. Given a distribution $\mathcal{P}$ over solutions $\mathcal{X}$, write $g(\mathcal{P}) = \max_{w \in \mathcal{W}} \mathbb{E}_{x \sim \mathcal{P}}[L(x, w)]$ for the worst-case expected loss of a solution drawn from $\mathcal{P}$. The goal of *improper robust optimization*: find a distribution $\mathcal{P}$ over solutions $\mathcal{X}$ such that $g(\mathcal{P}) \leq \alpha\tau$.

We will make the assumption that $L(x, w)$ is concave in $w$, 1-Lipschitz with respect to $w$ and that the set $\mathcal{W}$ is convex. The case of finite losses that we considered in the main text is a special case where the space $\mathcal{W}$ is the simplex on $m$ coordinates, and where: $L(x, w) = \sum_{i=1}^{m} w[i] \cdot L_i(x)$.

We will also assume that we are given access to an approximate stochastic oracle, which finds a solution $x \in \mathcal{X}$ that approximately minimizes a given distribution over loss functions:

**Definition 2** ($\alpha$-Approximate Stochastic Oracle). *Given a choice of $w \in \mathcal{W}$, the oracle $M(w)$ computes an $\alpha$-approximate solution $x^* = M(w)$ to the known parameter problem, i.e.:*

$$L(x^*, w) \leq \alpha \min_{x \in \mathcal{X}} L(x, w) \tag{23}$$

### B.1 Improper Robust Optimization with Oracles

We first show that, given access to an $\alpha$-approximate stochastic oracle, it is possible to efficiently implement improper $\alpha$-approximate robust optimization, subject to a vanishing additive loss term. The algorithm is a variant of Algorithm 1, where we replace the Multiplicative Weight Updates algorithm for the choice of $w_t$ with a projected gradient descent algorithm, which works for any convex set $\mathcal{W}$. To describe the algorithm we will need some notation. First we denote with $\Pi_{\mathcal{Y}}(w)$ to be the projection of $w$ on the set $\mathcal{Y}$, i.e. $\Pi_{\mathcal{W}}(w) = \arg\min_{w^* \in \mathcal{W}} \|w^* - w\|_2^2$. Moreover, $\nabla_y L(x, y)$ is the gradient of function $L(x, y)$ with respect to $y$.

**Theorem 7.** *Given access to an $\alpha$-approximate stochastic oracle, Algorithm 3, with $\eta = \frac{\max_{w \in \mathcal{W}} \|w\|_2}{\sqrt{2T}}$ computes a distribution $\mathcal{P}$ over solutions, defined as a uniform distribution over a set $\{x_1, \ldots, x_T\}$, such that:*

$$\max_{w \in \mathcal{W}} \mathbb{E}_{x \sim \mathcal{P}} [L(x, w)] \leq \alpha\tau + \max_{w \in \mathcal{W}} \|w\|_2 \sqrt{\frac{2}{T}} \tag{26}$$

*Proof.* We can interpret Algorithm 1 in the following way. We define a zero-sum game between a learner and an adversary. The learner's action set is equal to $\mathcal{X}$ and the adversaries action set is $W$.

The loss of the learner when he picks $x \in \mathcal{X}$ and the adversary picks $w \in \mathcal{W}$ is defined as $L(x, w)$. The corresponding payoff of the adversary is $L(x, w)$.

We will run no-regret dynamics on this zero-sum game, where at every iteration $t = 1, \ldots, T$, the adversary will pick a $w_t \in \mathcal{W}$ and subsequently the learner picks a solution $x_t$. We will be using the projected gradient descent algorithm to compute what $w_t$ is at each iteration, as defined in Equations (24) and (25). Subsequently the learner picks a solution $x_t$ that is the output of the $\alpha$-approximate stochastic oracle on the parameter chosen by the adversary at time-step $t$. That is,

$$x_t = M(w_t). \tag{27}$$

By the regret guarantees of the projected gradient descent algorithm for the adversary, we have that:

$$\frac{1}{T} \sum_{t=1}^{T} L(x_t, w_t) \geq \max_{w \in \mathcal{W}} \frac{1}{T} \sum_{t=1}^{T} L(x_t, w) - \epsilon(T) \tag{28}$$

for $\epsilon(T) = \max_{w \in \mathcal{W}} \|w\|_2 \sqrt{\frac{2}{T}}$. Combining the above with the guarantee of the stochastic oracle we have

$$
\begin{aligned}
\tau = \min_{x \in \mathcal{X}} \max_{w \in \mathcal{W}} L(x, w) &\geq \min_{x \in X} \frac{1}{T} \sum_{t=1}^{T} L(x, w_t) \\
&\geq \frac{1}{T} \sum_{t=1}^{T} \min_{x \in \mathcal{X}} L(x, w_t) \\
&\geq \frac{1}{T} \sum_{t=1}^{T} \frac{1}{\alpha} \cdot L(x_t, w_t) \qquad \text{(By oracle guarantee for each } t) \\
&\geq \frac{1}{\alpha} \cdot \left( \max_{w \in \mathcal{W}} \frac{1}{T} \sum_{t=1}^{T} L(x_t, w) - \epsilon(T) \right). \qquad \text{(By no-regret of adversary)}
\end{aligned}
$$

Thus if we define with $\mathcal{P}$ to be the uniform distribution over $\{x_1, \ldots, x_T\}$, then we have derived that

$$\max_{w \in \mathcal{W}} \mathbb{E}_{x \sim \mathcal{P}} [L(x, w)] \leq \alpha \tau + \epsilon(T) \tag{29}$$

as required. $\qquad \square$

A corollary of Theorem 7 is that if the solution space $\mathcal{X}$ is convex and the function $L(x, y)$ is also convex in $x$ for every $y$, then we can compute a single solution $x^*$ that is approximately minimax optimal.

**Corollary 8.** *If the space $\mathcal{X}$ is a convex space and the function $L(x, y)$ is convex in $x$ for any $y$, then the point $x^* = \frac{1}{T} \sum_{t=1}^{T} x_t \in \mathcal{X}$, where $\{x_1, \ldots, x_T\}$ are the output of Algorithm 3, satisfies:*

$$\max_{w \in \mathcal{W}} L(x^*, w) \leq \alpha \tau + \max_{w \in \mathcal{W}} \|w\|_2 \sqrt{\frac{2}{T}} \tag{30}$$

*Proof.* By Theorem 7, we get that if $\mathcal{P}$ is the uniform distribution over $\{x_1, \ldots, x_T\}$ then

$$\max_{w \in \mathcal{W}} \mathbb{E}_{x \sim \mathcal{P}}[L(x, w)] \leq \alpha \tau + \max_{w \in \mathcal{W}} \|w\|_2 \sqrt{\frac{2}{T}}.$$

Since $\mathcal{X}$ is convex, the solution $x^* = \mathbb{E}_{x \sim \mathcal{P}}[x]$ is also part of $\mathcal{X}$. Moreover, since each $L(x, y)$ is convex in $x$, we have that $\mathbb{E}_{x \sim \mathcal{P}}[L(x, y)] \geq L(\mathbb{E}_{x \sim \mathcal{P}}[x], y) = L(x^*, y)$. We therefore conclude

$$\max_{w \in \mathcal{W}} L(x^*, w) \leq \max_{w \in \mathcal{W}} \mathbb{E}_{x \sim \mathcal{P}}[L(x, w)] \leq \alpha \tau + \max_{w \in \mathcal{W}} \|w\|_2 \sqrt{\frac{2}{T}}$$

as required. $\qquad \square$

Our results for improper statistical learning can also be analogously generalized to this more general loss uncertainty.

## C  NP-Hardness of Proper Robust Optimization

The convexity assumption of Corollary 2 is necessary. In general, achieving any non-trivial ex-post robust solution is computationally infeasible, even when there are only polynomially many loss functions and they are all concave.

**Theorem 9.** *There exists a constant $c$ for which the following problem is NP-hard. Given a collection of linear loss functions $\mathcal{L} = \{\ell_1, \ldots, \ell_m\}$ over a ground set $N$ of $d$ elements, and an optimal stochastic oracle over feasibility set $\mathcal{X} = \{S \subset N : |S| = k\}$, find a solution $x^* \in \mathcal{X}$ such that*

$$\max_{\ell \in \mathcal{L}} \ell(x^*) \leq \tau + \frac{k}{m}.$$

*Proof.* We reduce from the set packing problem, in which there is a collection of sets $\{T_1, \ldots, T_d\}$ over a ground set $\mathcal{U}$ of $m$ elements $\{u_1, \ldots, u_m\}$, and the goal is to find a collection of $k$ sets that are all pairwise disjoint. This problem is known to be NP-hard, even if we assume $k < m/4$.

Given an instance of the set packing problem, we define an instance of robust loss minimization as follows. There is a collection of $m$ linear functions $\mathcal{L} = \{\ell_1, \ldots, \ell_m\}$, and $N$ is a set of $mk + d$ items, say $\{a_{ij}\}_{i \leq m, j \leq k} \cup \{b_r\}_{r \leq d}$. The linear functions are given by $\ell_i(a_{ij}) = 1/k$ for all $i$ and $j$, $\ell_i(a_{i'j}) = 0$ for all $i' \neq i$ and all $j$, $\ell_i(b_r) = 2/m$ if $u_i \in T_r$, and $\ell_i(b_r) = 1/km$ if $u_i \notin T_r$.

We claim that in this setting, an optimal stochastic oracle can be implemented in polynomial time. Indeed, let $D$ be any distribution over $\mathcal{L}$, and let $\ell_i$ be any function with minimum probability under $D$. Then the set $S = \{a_{i1}, \ldots, a_{ik}\}$ minimizes the expected loss under $D$. This is because the contribution of any given element $a_{ij}$ to the loss is equal to $1/k$ times the probability of $\ell_i$ under $D$, which is at most $1/km$ for the lowest-probability element, whereas the loss due to any element $b_r$ is at least $1/km$. Thus, since the optimal stochastic oracle is polytime implementable, it suffices to show NP-hardness without access to such an oracle.

To establish hardness, note that if a set packing exists, then the solution to the robust optimization problem given by $S = \{b_r : T_r \text{ is in the packing }\}$ satisfies $\ell_i(S) \leq 2/m + (k-1)/km < 3/m$. On the other hand, if a set packing does not exist, then any solution $S$ for the robust optimization problem either contains an element $a_{ij}$ — in which case $\ell_i(S) \geq 1/k > 4/m$ — or must contain at least two elements $b_r, b_s$ such that $T_r \cap T_s \neq \emptyset$, which implies there exists some $i$ such that $\ell_i(S) \geq 4/m$. We can therefore reduce the set packing problem to the problem of determining whether the minimax optimum $\tau$ is greater than $4/m$ or less than $3/m$. We conclude that it is NP-hard to find any $S^*$ such that $\max_{\ell \in \mathcal{L}} \ell(S^*) \leq \tau + 1/m$. $\qquad\square$

Similarly, for robust submodular maximization, in order to achieve a non-trivial approximation guarantee it is necessary to either convexify the outcome space (e.g., by returning distributions over solutions) or extend the solution space to allow solutions that are larger by a factor of $\Omega(\log |\mathcal{F}|)$. This is true even when there are only polynomially many functions to optimize over, and even when they are all linear.

**Theorem 10.** *There exists a constant $c$ for which the following problem is NP-hard. Given any $\alpha > 0$, and a collection of linear functions $\mathcal{F} = \{f_1, \ldots, f_m\}$ over a ground set $N$ of $d$ elements, and an optimal stochastic oracle over subsets of $N$ of size $k$, find a subset $S^* \subseteq N$ with $|S^*| \leq ck \log(m)$ such that*

$$\min_{f \in \mathcal{F}} f(S^*) \geq \frac{1}{\alpha}\tau - \frac{1}{\alpha km}.$$

*Proof.* We reduce from the set cover problem, in which there is a collection of sets $\{T_1, \ldots, T_d\}$ over a ground set $\mathcal{U}$ of $m$ elements $\{u_1, \ldots, u_m\}$, whose union is $\mathcal{U}$, and the goal is to find a collection of at most $k$ sets whose union is $\mathcal{U}$. There exists a constant $c$ such that it is NP-hard to distinguish between the case where such a collection exists, and no collection of size at most $ck \log(n)$ exists.

Given an instance of the set cover problem, we define an instance of the robust linear maximization problem as follows. There is a collection of $m$ linear functions $\mathcal{F} = \{f_1, \ldots, f_m\}$, and $N$ is a set of $km + d$ items, say $\{a_{ij}\}_{i \leq m, j \leq k} \cup \{b_r\}_{r \leq d}$. For each $i \leq m$ and $j \leq k$, set $f_i(a_{ij}) = 1/k$ and $f_i(a_{i'j}) = 0$ for all $i' \neq i$. For each $i \leq m$ and $r \leq d$, set $f_i(b_r) = 1/km$ if $u_i \in T_r$ in our instance of the set cover problem, and $f_i(b_r) = 0$ otherwise.

We claim that in this setting, an optimal stochastic oracle can be implemented in polynomial time. Indeed, let $D$ be any distribution over $\mathcal{F}$, and suppose $f_i$ is any function with maximum probability under $D$. Then the set $S = \{a_{i1}, \ldots, a_{ik}\}$ maximizes expected value under $D$. This is because the value of any given element $a_{ij}$ is at least $1/k$ times the probability of $f_i$ under $D$, which is at least $1/m$, whereas the value of any element $b_r$ is at most $1/km$. Thus, since the optimal stochastic oracle is polytime implementable, it suffices to show NP-hardness without access to such an oracle.

To establish hardness, note first that if a solution to the set cover problem exists, then the solution to the robust optimization problem given by $S = \{b_r : T_r$ is in the cover $\}$ satisfies $f_i(S) \geq 1/km$ for all $i$. On the other hand, if no set cover of size $k$ exists, then for any solution $S$ to the robust optimization problem there must exist some element $u_i$ such that $u_i \neq T_r$ for every $b_r \in S$, and such that $a_{ij} \neq S$ for all $j$. This implies that $f_i(S) = 0$, and hence $\tau = 0$. We have therefore reduced the set cover problem to distinguishing cases where $\tau \geq 1/km$ from cases where $\tau = 0$. We conclude that it is NP-hard to find any $S^*$ for which $\min_{f \in F} f(S^*) \geq \frac{1}{\alpha}(\tau - \frac{1}{km})$, for any positive $\alpha$. $\qquad\square$

# D  Strengthening the Benchmark

We now observe that our construction actually competes with a stronger benchmark than $\tau$. In particular, one that allows for distributions over solutions:

$$\tau^* = \min_{\mathcal{G} \in \Delta(\mathcal{X})} \max_{i \in [m]} \mathbb{E}_{x \sim \mathcal{G}}[L_i(x)] \tag{31}$$

Hence, our assumption is that there exists a distribution $\mathcal{G}$ over solutions $\mathcal{X}$ such that for any realization of the objective function, the expected value of the objective under this distribution over solutions is at least $\tau^*$.

Now we ask: given an oracle for the distributional problem, can we find a solution for the robust problem that achieve minimum reward at least $\tau^*$. We show that this is possible:

**Theorem 11.** *Given access to an $\alpha$-approximate stochastic oracle, we can compute a distribution $\mathcal{P}$ over solutions, defined as a uniform distribution over a set $\{x_1, \ldots, x_T\}$, such that:*

$$\max_{i \in [m]} \mathbb{E}_{x \sim \mathcal{P}}[L_i(x)] \leq \alpha \tau^* + \sqrt{\frac{2 \log(m)}{T}} \tag{32}$$

*Proof.* Observe that a stochastic oracle for the setting with solution space $\mathcal{X}$ and functions $\mathcal{L} = \{L_1, \ldots, L_m\}$ is also a stochastic oracle for the setting with solution space $\mathcal{D} = \Delta(\mathcal{X})$ and functions $\mathcal{L}' = \{L'_1, \ldots, L'_m\}$, where for any $D \in \mathcal{D}$: $L'_j(D) = \mathbb{E}_{x \sim D}[L_j(x)]$. Moreover, observe that $\tau^*$ is exactly equal to $\tau$ for the setting with solution space $\mathcal{D}$ and function space $\mathcal{L}'$. Thus applying Theorem 1 to that setting we get an algorithm which computes a distribution $\mathcal{P}'$ over distributions of solutions in $\mathcal{X}$, that satisfies:

$$\max_{i \in [m]} \mathbb{E}_{D \sim \mathcal{P}'}[\mathbb{E}_{x \sim D}[L_j(x)]] \leq \alpha \tau^* + \sqrt{\frac{2 \log(m)}{T}} \tag{33}$$

Observe that a distribution over distributions of solutions is simply a distribution over solutions, which concludes the proof of the Theorem. $\qquad\square$

# E  Experiments

## E.1  Hybrid Method

In order to apply the robust optimization algorithm we need to construct a neural network architecture that facilitates it. In each iteration $t$, such an architecture receives a distribution over corruption types $\mathbf{w}_t = [\mathbf{w}_t[1], ..., \mathbf{w}_t[m]]$ and produces a set of weights $\theta_t$.

Figure 4: First interpretation of stochastic oracle, training on a sample of images drawn from the mixture of corruptions.

In the Hybrid Method, our first oracle, we take each training data image and perturb it by exactly one corruption, with corruption $i$ being selected with probability $\mathbf{w}_t[i]$. Then apply mini-batch gradient descent, picking mini-batches from the perturbed data set, to train a classifier $\theta_t$. Note that the resulting classifier will take into account corruption $i$ more when $\mathbf{w}_t[i]$ is larger.

## E.2 Composite Method

Figure 5: Second interpretation of stochastic oracle, by creating $m$ coupled instantiations of the net architecture (one for each corruption type), with the $i$-th instance taking as input the image corrupted with the $i$-th corruption and then defining the loss as the convex combination of the losses from each instance.

In the Composite Method, at each iteration, we use $m$ copies of the training data, where copy $i$ has Corruption Type $i$ applied to all training images. The new neural network architecture has $m$ sub-networks, each taking in one of the $m$ training data copies as input. All sub-networks share the same set of neural network weights. During a step of neural network training, a mini-batch is selected from the original training image set, and the corresponding images in each of the $m$ training set copies are used to compute weighted average of the losses $\sum_{i=1}^{m} \mathbf{w}_t[i] Loss_{t,i}$, which is then used to train the weights.

## E.3 Corruption Set Details

**Background Corruption Set** consists of images with (i) an unperturbed white background–the original images, (ii) a light gray tint background, (iii) a gradient background, (iv) and a checkerboard background.

**Shrink Corruption Set** consists of images with (i) no distortion–the original images, (ii) a 25% shrinkage along the horizontal axis, (iii) a 25% shrinkage along the vertical axis, and (iv) a 25% shrinkage in both axes.

**Pixel Corruption Set** consists of images that (i) remain unaltered–the original images, (ii) have $Unif[-0.15, -0.05]$ perturbation added i.i.d. to each pixel, (iii) have $Unif[-0.05, 0.05]$ perturbation added i.i.d. to each pixel, and (iv) have $Unif[0.05, 0.15]$ perturbation added i.i.d. to each pixel.

**Mixed Corruption Set** consists of images that (i) remain unaltered–the original images, and one corruption type from each of the previous three corruption sets (which were selected at random), namely that with (ii) the checkerboard background, (iii) 25% shrinkage in both axes, and (iv) i.i.d. $Unif[-0.15, -0.05]$ perturbation.

## E.4   Neural Network Results

|  | *Background Set* | *Shrink Set* | *Pixel Set* | *Mixed Set* |
|---|---|---|---|---|
| *Best Individual Baseline* | **8.85** | **7.19** | **1.82** | **8.75** |
|  | *(8.38,9.32)* | *(7.09,7.28)* | *(1.81,1.82)* | *(8.50,9.00)* |
| *Even Split Baseline* | **28.35** | **11.54** | **1.93** | **9.92** |
|  | *(26.81,29.89)* | *(11.25,11.83)* | *(1.91,1.95)* | *(9.78,10.06)* |
| *Uniform Distribution Baseline* | **2.06** | **1.74** | **1.30** | **1.46** |
|  | *(2.05,2.08)* | *(1.72,1.76)* | *(1.30,1.31)* | *(1.45,1.47)* |
| *Hybrid Method* | **1.38** | **1.48** | **1.29** | **1.36** |
|  | *(1.37,1.39)* | *(1.47,1.49)* | *(1.28,1.30)* | *(1.35,1.36)* |
| *Composite Method* | **1.31** | **1.30** | **1.25** | **1.25** |
|  | *(1.30,1.31)* | *(1.29,1.31)* | *(1.24,1.25)* | *(1.24,1.26)* |

Table 1: Individual Bottleneck Loss results (mean over 10 independent runs and a 95% confidence interval for the mean) with $T = 50$ on all four Corruption Sets. Composite Method outperforms Hybrid Method, and both outperform baselines, with such differences being statistically significant.

## E.5   Analysis of Multiplicative Weights Update

Consider the robust optimization algorithm using the Hybrid and Composite Methods, but parameterizing $\eta$ as $\eta = c \cdot T^{-\gamma}$ (for constant $c = \sqrt{\frac{\log m}{2}}$) to alter the multiplicative weights update formula. In this paper, we have been using $\gamma = 0.5 \implies \eta = \frac{c}{\sqrt{T}}$. Lower values of $\gamma$ leads to larger changes in the distribution over corruption types between robust optimization iterations. Here we rerun our experiments from Section 3.1 using $\gamma = 0.1$; we did not tune $\gamma$–the only values of $\gamma$ tested were 0.1 and 0.5.[5]

Figure 6: Comparison of Individual Bottleneck Loss between using $\gamma = 0.5$ vs. $\gamma = 0.1$ in the multiplicative weights update, for both the Hybrid and Composite Methods. The $\gamma = 0.1$ setting yields lower loss.

The improved performance with $\gamma = 0.1$ compared to $\gamma = 0.5$ is related to an important property of our robust optimization algorithm in practice–namely that $\mathbf{w}$ stabilizes for sufficiently large $T$. Over the course of iterations of the algorithm, $\mathbf{w}$ moves from the initial discrete uniform distribution to some optimal *stable distribution*, where the stable distribution is consistent across independent runs. The $\gamma = 0.1$ setting yields to better Individual Bottleneck Loss than the $\gamma = 0.5$ setting for finite $T$ because it converges more rapidly to the stable distribution.

Figure 7: **Left:** The amount that the distribution over corruption types $\mathbf{w}$ changes between iteration $t$ & $t+1$ decays rapidly as $t$ increases, and the distribution stabilizes. Plot shows 16 time series, corresponding to results for each combination of ({Hybrid, Composite},{$\gamma = 0.5, \gamma = 0.1$},{Background, Shrink, Pixel, Mixed}), using the mean over 10 runs. **Right:** The difference between $\gamma = 0.1$ & $\gamma = 0.5$ in the amount that $\mathbf{w}$ changes between iterations. Shows the difference between pairs of time series from the previous figure (thus there are $\frac{16}{2} = 8$ time series shown). Values are positive for small $t$ and near 0 for larger $t$, showing that the $\gamma = 0.1$ setting yields faster changes in $\mathbf{w}$ initially, thereby allowing $\mathbf{w}$ to more quickly approach the stable distribution.

# F  Experiments on Robust Influence Maximization

## F.1  Influence Results

|  | Wikipedia A | Wikipedia B | Complete A | Complete B |
|---|---|---|---|---|
| *Individual Baseline* | **56.56** | **35.84** | **19.77** | **11.27** |
|  | *(53.55,59.57)* | *(31.93,39.75)* | *(16.57,22.96)* | *(10.77,11.77)* |
| *Uniform Baseline* | **82.30** | **46.60** | **3.10** | **5.20** |
|  | *(78.19,86.41)* | *(40.53,52.67)* | *(2.24,3.96)* | *(4.07,6.33)* |
| *Perturbed Dist. Baseline* | **83.35** | **48.92** | **21.99** | **10.14** |
|  | *(79.87,86.82)* | *(43.80,54.03)* | *(17.38,26.61)* | *(9.37,10.91)* |
| *Robust Optimization* | **94.33** | **66.42** | **36.34** | **17.91** |
|  | *(90.61,98.05)* | *(64.17,68.66)* | *(33.46,39.21)* | *(17.22,18.60)* |

Table 2: Mean worst-case influence $\min_{i \in [m]} E_{S \sim \mathcal{P}}[f_i(S)]$ for the solution $\mathcal{P}$ returned by each method, over 10 independent runs using $T = 200$, and 95% confidence intervals for those means.

Robust Optimization outperforms the baselines, and the differences are statistically significant.[6]

## F.2  Performance of Single Solutions

For the Complete Graph $A$ case, it is computationally feasible to obtain the absolute best seed set (via brute force over $\binom{100}{2}$ total possible seed sets), so we can consider the ratio of the best individual seed set generated at some iteration $t$ by robust optimization to the absolute best seed set–that is, $\frac{\max_{S \in \mathcal{P}} \min_{i \in [m]} f_i(S)}{\max_S \min_{i \in [m]} f_i(S)}$. The mean of this ratio over 10 runs was 0.733.

For the other three cases, it is not computationally feasible to obtain the absolute best seed set, but we can instead compare the best individual seed set generated by the robust optimization procedure to the Bottleneck Influence value from considering all of $\mathcal{P} = \{S_1, ..., S_T\}$–specifically, the ratio $\frac{\max_{S \in \mathcal{P}} \min_{i \in [m]} f_i(S)}{\min_{i \in [m]} E_{S \sim \mathcal{P}} f_i(S)}$. Based on the mean of 10 runs, this ratio is 0.995 for Wikipedia $A$, 0.855 for Wikipedia $B$, and 0.509 for Complete $B$. The individual seed sets generated by the robust optimization procedure are thus especially good for the Wikipedia Graph; those Wikipedia Graph results are more representative of real graphs, since the Complete Graph has an artificially small number of nodes ($|V| = 100$).

## Footnotes

[5]A possible future step would be to use cross-validation to tune $\gamma$ or design an adaptive parameter algorithm for $\gamma$.

[6]Claim of statistical significance is based on means of differences between methods, which controls for differences in the $G_i$, rather than differences between means, which are shown in Table 2.