[Reviews · NeurIPS 2017]

Reviewer 1



The paper discusses approximate robust optimization over a finite set of loss functions (of size m), which are possibly nonconvex. It is shown that the proposed algorithm provides a distribution over a set of solutions (of size T) that achieves a theoretical bound of order sqrt(log(m)/T) on the distance to an alpha-approximate solution of the robust optimization problem. The results apply for a single solution point rather than a distribution, when the loss functions and the solution space are convex. Two application examples are presented and employed for the numerical testing. However, I believe that there are a few points that needs clarification. 1. The contribution over the work in reference [17] should be clearly stated as the setup looks very similar. As fas as I can see Theorem 1 in [17] applies to general loss functions, with an error bound on subproblem minimization. Thus it is important to stress the distinction. 2. It is not clear to me whether Algorithm 1 and Definition 1 exactly correspond to the terms 'Bayesian optimization' and 'Bayesian oracle' used in relevant literature. It would be good to have a short explanatory paragraph if so, and change the terminology otherwise.

Reviewer 2



This study showed that a minimax problem can be approximately solved via approximate Bayesian oracle (Theorem 1). Overall, the study has a solid contribution. I checked all the proofs and did not found a serious flaw. Even the technique is not very involved, it has wide applications. [Concern] Theorem 1 seems a standard argument that converts min-max and min-sum, which is sometimes used to prove the minimax theorem via no-regret learning. (I would like to hear a key-point of the proof that I missed) The title is misleading. First, it does not tackle the non-convexity (the difficulty of non-convex problem is the hardness of constructing an approximation oracle; thus the existence of the oracle is too strong assumption for this title). Second, it is not limited for continuous problems ("non-convex" is usually used for continuous problem, but the only (theoretically solid) application is a discrete problem). I guess the authors want to apply the method to train the neural network (as they did in the experiment) so the paper has this title. But I think it is not a good title. [Minor] line 155 and 156, "\subseteq 2^V" should be "\subseteq V" or "\in 2^V".

Reviewer 3



Summary: This work provides a method to optimize the worst-case loss over a set of different class of non-convex objective functions. In the real world, observed signals may contain various source of noise, which requires the learning method to be robust enough to the noise. The author converted the robust improper optimization to a Bayesian optimization problem. The proposed learning procedure consists of two iterative steps: the learner tries to minimize the loss with an alpha-approximate Bayesian oracle; the adversary aims to find a distribution over losses that maximize the loss of the learner. The author provide detailed proofs for the convergence and the theoretical bounds of the algorithm. Specifically, a robust statistical learning and a robust submodular maximization cases were discussed. Supportive application experiments were performed to validate the method. The result shows that the proposed method has substantial improvements in the worst-case scores. Quality: The paper is well written with theoretical proofs and supportive experiments. The author compared the proposed method with many modifications to support their choice. Clarity: The paper is well organized. The experiment details are clearly written. Originality: There are a lot of works in robust optimization in machine learning. This work focuses on more general setting of robust optimization where loss functions are non-convex. The application of alpha-approximate Bayesian oracle is a novel approach. Significance: This work provides a general approach for robust optimization of non-convex objective functions. The method is validated from both a neural network learning task and an influence maximization task and it substantially outperformed the baseline approaches. It leads the state-of-art in robust optimization for non-convex problems.